# Lignin-polysaccharide interactions in plant secondary cell walls revealed by solid-state NMR

Xue Kang [1], Alex Kirui [1], Malitha C. Dickwella Widanage [1], Frederic Mentink-Vigier [2], Daniel J. Cosgrove[3] & Tuo Wang [1]

Lignin is a complex aromatic biopolymer that strengthens and waterproofs plant secondary cell walls, enabling mechanical stability in trees and long-distance water transport in xylem. Lignin removal is a key step in paper production and biomass conversion to biofuels, motivating efforts to re-engineer lignin biosynthesis. However, the physical nature of lignin's interactions with wall polysaccharides is not well understood. Here we show that lignin self-aggregates to form highly hydrophobic and dynamically unique nanodomains, with extensive surface contacts to xylan. Solid-state NMR spectroscopy of intact maize stems, supported by dynamic nuclear polarization, reveals that lignin has abundant electrostatic interactions with the polar motifs of xylan. Lignin preferentially binds xylans with 3-fold or distorted 2-fold helical screw conformations, indicative of xylans not closely associated with cellulose. These findings advance our knowledge of the molecular-level organization of lignocellulosic biomass, providing the structural foundation for optimization of post-harvest processing for biofuels and biomaterials.

[1] Department of Chemistry, Louisiana State University, Baton Rouge, LA 70803, USA. [2] National High Magnetic Field Laboratory, Tallahassee, FL 32310, USA. [3] Department of Biology, Pennsylvania State University, University Park, PA 16802, USA. These authors contributed equally: Xue Kang, Alex Kirui. Correspondence and requests for materials should be addressed to T.W. (email: tuowang@lsu.edu)

The secondary cell wall comprises the majority of plant biomass and is a sophisticated composite of cellulose, hemicellulose (mainly xylan and glucomannan) and lignin[1,2]. This supramolecular network formed by complex carbohydrates and aromatic polymers provides the cell with sufficient mechanical strength and rigidity, but it also makes the lignocellulosic materials inherently recalcitrant to chemical and enzymatical treatments during biofuel production[3]. Decades of efforts have been devoted to genetically engineering the plants to improve the composition and structure of cell wall polymers, aiming at increased digestibility[4–6]. This method offers the potential for effectively generating sustainable bioenergy, however, a major hurdle here is our inadequate understanding of the cell wall architecture on the molecular level.

In secondary cell walls of woody materials, bundles of cellulose microfibrils, typically 10–20 nm across[7], are proposed to be coated by a xylan-lignin complex and crosslinked by glucomannans[8–11]. However, due to the inherent, technical constraints of traditional analytical methods, detailed molecular information about secondary cell wall organization has remained scarce. Conventional methods either rely on sequential extractions followed by compositional analysis or require partial dissolution of the lignocellulosic materials using organic solvents and ionic liquids for solution-NMR characterization[12–14]. These procedures substantially perturb the physical state and molecular interactions of biomolecules, introducing considerable uncertainty to our understanding of the cell wall structure. This limitation has been partially alleviated by the recent solid-state NMR (ssNMR) work that characterized xylan polymorphism in native secondary cell walls of *Arabidopsis*, revealing a two-fold helical screw symmetry with a regular pattern of acetate or glucuronate substitutions in cellulose-bound xylan[15–17]. However, many structural aspects still await investigation and a key question, examined here, is how lignin and polysaccharides are packed in intact secondary cell walls.

The atomic resolution of ssNMR spectroscopy and the sensitivity enhancement provided by the dynamic nuclear polarization (DNP) technique have enabled us to clarify and substantiate our ambiguous view of lignocellulose structure with detailed molecular evidence. The $^{13}$C-labeled stems from three energy and agricultural crops (maize, rice, switchgrass) as well as the model plant *Arabidopsis* were investigated. Dominant interactions between xylan with non-flatten conformations and lignin units rich in methyl ethers are observed whereas direct lignin-cellulose interactions are less prominent. Because the degree of hydration and timescale of motions are distinct between the lignin and polysaccharides, we propose that lignin self-aggregates in distinctive nanodomains with extensive surface contacts to hemicelluloses. These results provide a substantial revision of our understanding of the supramolecular architecture of secondary plant cell walls, which can facilitate the development of crops with higher digestibility and improve the efficiency of biomass deconstruction and conversion to biofuels.

## Results

### Polysaccharides and lignin are structurally polymorphic.
Uniformly $^{13}$C-labeled stems of maize, rice, switchgrass, and *Arabidopsis* were produced for ssNMR analysis by growing the plants in a closed growth chamber with continuous supply of $^{13}CO_2$. Isotopic enrichment provides adequate sensitivity for systematically measuring two-dimensional (2D) $^{13}$C–$^{13}$C spectra, which provide atomic resolution for determining the composition, sub-nanometer packing, site-specific hydration, and ns-µs motion of lignin and polysaccharides in the near-native state.

Figure 1 shows representative 2D $^{13}$C–$^{13}$C correlation INADEQUATE spectra and molecular structures of biomolecules

in intact maize stems. These samples contained predominantly secondary cell walls, as xyloglucan (a primary wall component) was negligible (Fig. 1a). The major hemicellulose, xylan, is dominated by its 2-fold, extended conformers (Xn$^{2f}$), accounting for ~70 mol% as indicated by the peak volume (Fig. 1b). This is potentially due to the extensive interaction of xylan with cellulose microfibrils, which promotes the two-fold, flat conformation. The interactions with cellulose rigidify the two-fold xylan and further enhance its signal in the current spectrum that relies on the $^{1}$H-$^{13}$C cross-polarization (CP) to preferentially detect rigid components. These two-fold xylans are structurally polymorphic (Fig. 1a) and only a subgroup is in close contact with lignin as shown later. The 2- and three-fold xylan conformers are both heavily acetylated and mixed on the sub-nanometer scale because a moderate cross peak exists between the C1 of two-fold xylan and the C4 of the three-fold conformer (Xn1$^{2f}$-Xn4$^{3f}$) (Supplementary Fig. 1a).

The glucan chains in cellulose microfibrils of secondary cell walls are highly polymorphic in structure. Among the six major allomorphs, types a–d are the internal chains while types g and f originate from surface residues (Fig. 1a). With the high resolution attained here, NMR further reveals subtle differences in cellulose organization in cell walls from growing coleoptiles versus mature stems (Fig. 1c) where microfibrils aggregate extensively[7,18]. The increase in relative intensities of the deeply embedded allomorph (type-c) and the reduced number of surface chains (Fig. 1c and Supplementary Table 1) collectively indicate the restructuring of elementary microfibrils on the aggregation interface, likely enabled by cooperative activity of multiple cellulose synthase complexes during secondary wall formation[19,20].

Maize lignin is mainly composed of *p*-hydroxyphenyl (H), guaiacyl (G), syringyl (S) and ferulate (FA) residues (Fig. 1e–g). The H, G, and S lignin units differ in the number of methoxyl substitutions at carbon 3 and 5, which results in well-resolved signals for the ring carbons. These aromatics bear conformational heterogeneity since they show broad linewidth and peak multiplicity (Supplementary Fig. 1). The $^{13}$C NMR chemical shifts are documented in Supplementary Table 2, and the well-resolved signals of aromatics and carbohydrates enable further determination of their intermolecular packing as detailed below.

### Lignin binds xylan through electrostatic interactions.
Both native and modified lignins have been extensively studied using solution NMR[13,14,21,22], but native interactions with other wall components are lost in the ball-milled, fractioned, and dissolved samples. Using the intact maize stems, we have measured a long-mixing (1.0 s) 2D spectrum, which displays 74 intermolecular cross peaks that are absent in the 0.1-s short-range correlation spectrum (Fig. 2a). This experiment provides exquisite details on the spatial proximities (instead of covalent bonding) of cell wall polymers as each cross peak represents a unique sub-nanometer contact between two distinct atoms in adjacent molecules. These intermolecular interactions are well-resolved in the NMR spectra and can be classified into four major categories depending on the interaction site and the structural motifs. First, cross peaks may occur between two different lignin units, for instance, the H4-S3/5 cross peak at (159, 153 ppm). Second, the methyl ethers of lignin may be spatially proximal to xylan and cellulose. Notable examples include the OMe-s4 at (57, 84 ppm) and the OMe-Xn4$^{3f}$ at (57, 78 ppm) caused by the contacts between lignin methyl ethers and surface cellulose carbon 4 and three-fold xylan carbon 4. Third, the aromatic carbons of lignin may correlate with the ring carbons of carbohydrates, for example, the S3/5-Xn4$^{3f}$ cross peak at (153, 78 ppm). Fourth, the xylan acetyl group may correlate with lignin as reflected by the Ac$^{Me}$-FA9 cross peak at (21, 169 ppm) and the Ac$^{Me}$-S3/5 cross peak at (21, 153 ppm),

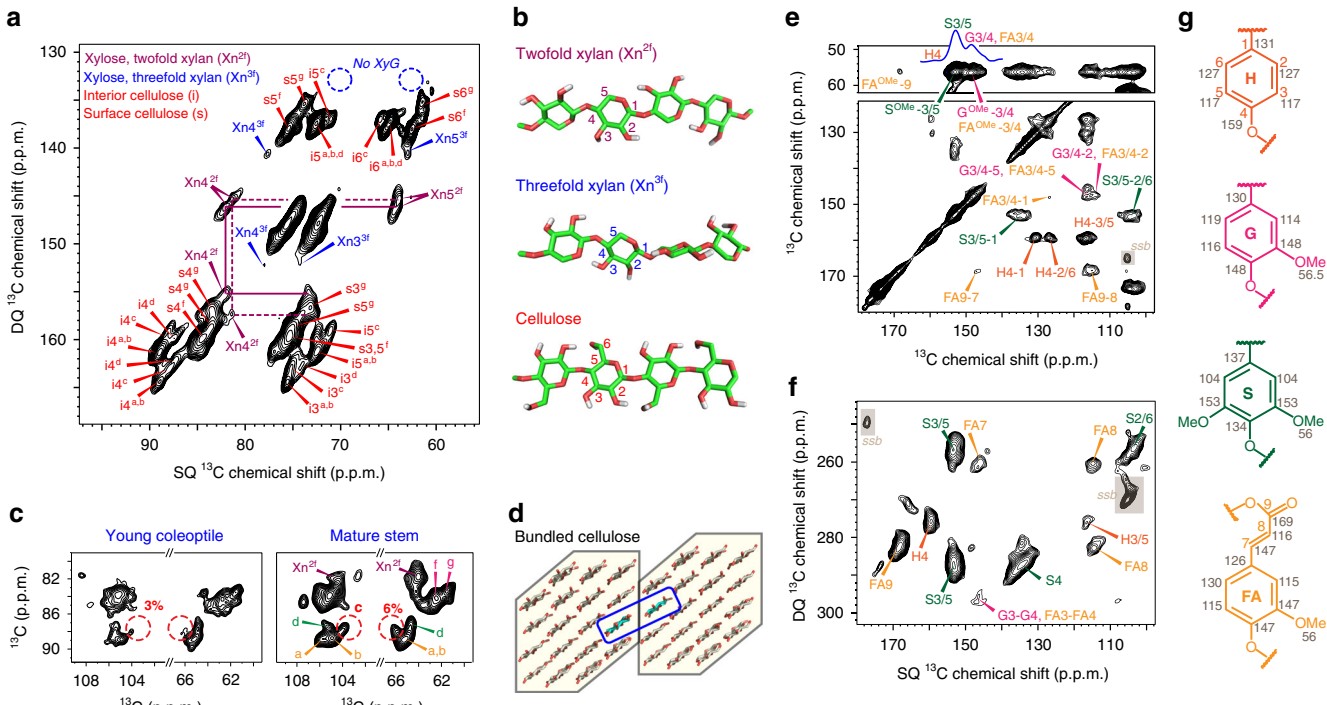

**Fig. 1** The polymorphic structure of lignin and polysaccharides in intact maize stems. **a** Representative 2D $^{13}$C–$^{13}$C correlation spectrum resolves the polymorphic signals of cellulose and xylan. The xyloglucan (XyG) signals are missing (blue circles), indicating a negligible amount of primary cell walls. Abbreviations are used for assignment, e.g., s4$^g$ is carbon 4 of glucose type-g on the microfibril surface. **b** Representative polysaccharide structures. **c** Cellulose signals in maize coleoptile and mature stems. The six cellulose allomorphs are labeled using letters **a–g**. The intensity of type-c cellulose is 2X greater in mature stems compared with coleoptiles. **d** An illustrative figure of two adjacent microfibrils, with the restructured chains boxed in blue, which increase the intensity of type-c cellulose in **c**. These chains restructure by changing the hydroxymethyl conformation. The number of glucan chains in the figure may not represent the actual microfibril structure. The aromatic signals of lignin are resolved using 2D $^{13}$C–$^{13}$C spectra measured using **e** 53-ms CORD and **f** INADEQUATE methods. **g** Representative structures and $^{13}$C chemical shifts of lignin

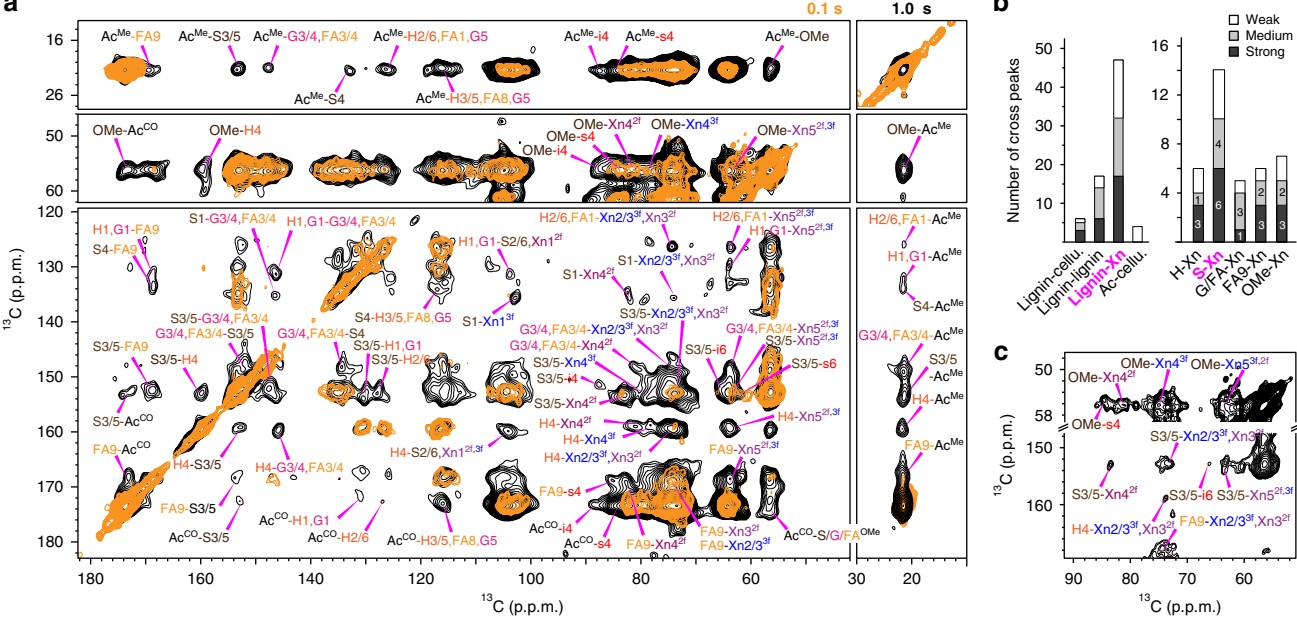

**Fig. 2** Electrostatic interactions of xylan and S units dominate lignin-polysaccharide interactions in maize. **a** Representative $^{13}$C–$^{13}$C spectra measured with short (0.1 s, orange) and long (1 s, black) mixing times on maize. The 1-s spectrum shows many long-range intermolecular cross peaks that are absent in the 0.1-s spectrum. Only intermolecular cross peaks are labeled. **b** Summary of the 74 intermolecular cross peaks in maize. For each category, the peaks are grouped by their strength. Cellulose is abbreviated as cellu. The interaction types with the most cross peaks are highlighted in magenta. Lignin-xylan contacts dominate, with the most cross peaks for S. Source Data are provided as a Source Data file. **c** The Xn-S and Xn-OMe cross peaks in the 100-ms spectrum reveal close contacts

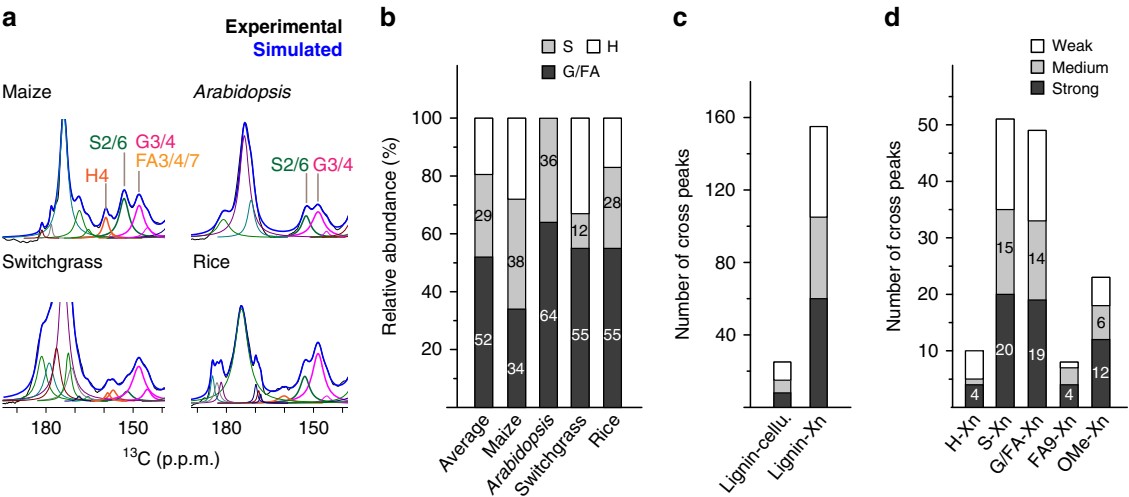

**Fig. 3** Lignin composition and carbohydrate interactions in four species. **a** Spectral deconvolution of quantitative $^{13}$C spectra for compositional analysis of lignin. **b** The molar composition of lignin from different species. Note that the FA/G is only G in *Arabidopsis*. **c** Summary of lignin-carbohydrate interactions. Lignin mainly interacts with xylan instead of cellulose. **d** The 155 xylan-lignin cross peaks categorized by the lignin type and peak intensities. Source Data are provided as a Source Data file for Fig. 3b–d

which originate from physical contacts between the methyl carbon of xylan acetyl groups with the carbon 9 of ferulate and the carbon 3/5 of syringyl units.

All intermolecular correlations are categorized by the interacted molecules and the relative intensities of these cross peaks (Fig. 2b). Xylan is the primary polysaccharide interactor of lignin with 47 cross peaks (Fig. 2b) and tightly contacts S-lignin (<5 Å), evidenced by cross peaks with only a short mixing period (Fig. 2c). Although cellulose-lignin contacts have been proposed in-silico[23], the scarcity of cross peaks indicate few direct interactions in these non-woody samples (Fig. 2b).

We have extended these experiments to two other economically important grasses, switchgrass and rice, and to *Arabidopsis*, a model dicot (Fig. 3 and Supplementary Fig. 2). The lignin composition varies substantially among species: maize contains all the four units; in *Arabidopsis* S and G predominate; switchgrass lacks S and rice lacks H (Fig. 3a, b and Supplementary Fig. 1c). In these plants, we have identified 234 distinctive intermolecular cross peaks, and the large number of correlations identified here allows us to conduct statistical analysis of polymer contacts in this collection of plant species. Lignin has 155 cross peaks with xylan but only 25 contacts with cellulose, confirming the anchoring role of xylan (Fig. 3c). Despite its modest abundance (29% on average) (Fig. 3b), the S-residue has the most abundant cross peaks with xylan (Fig. 3d). The amount of G and FA (52%) is almost twice that of S but their xylan cross peaks are less prevalent. The number of physical contacts with polysaccharide correlates with the number of methyl ether groups (OMe) in lignin and decreases in the order: S > G/FA > H. Consistently, 80% of the OMe-Xn cross peaks are either strong or medium (Fig. 3d), indicating that electrostatic interactions between lignin methoxy groups and xylan polar functionalities dominate their physical contacts. The strong polysaccharide interaction of S units may be related to its early deposition during lignification[24] and the weak H-Xn interaction enlightens the higher lignin extractability in H-rich plants[25].

**Xylan with a non-flatten conformation binds lignin.** The lignin-binding capacity of xylan is found to be conformation-dependent. We measured a lignin-edited spectrum to selectively probe the interface between lignin and polysaccharide (Supplementary Fig. 3, 4). This challenging experiment is enabled by a

23-fold sensitivity enhancement (Fig. 4a) that is achieved using the DNP technique, which transfers polarization from electrons to NMR-active nuclei[26–29]. The lignin-bound polysaccharides contain xylan and part of the cellulose surface (Fig. 4b). In these maize walls, xylan in three-fold conformation constitutes less than one-third of all xylans but accounts for almost half of the lignin-bound xylans; therefore, three-fold conformation facilitates xylan-aromatic interactions. In contrast, the two-fold xylans are structurally polymorphic and lignin only binds subtypes b and c, whose signals are sandwiched between the flat conformer (type-a) and the three-fold allomorph. Since the xylan $^{13}$C4 chemical shift indicates glycosidic bond conformation, it shows that the flat-ribbon structure of the two-fold conformation is not favorable for binding the intrinsically disordered lignin polymer.

**Lignin self-aggregates to form a highly hydrophobic domain.** To probe the hydration profile of secondary cell walls, we have compared the intensities of water-associated molecules with those of the whole cell wall (Supplementary Fig. 5). The relative intensities correlate with the extent of water retention and reveal a decreasing hydration gradient in the order: 3-fold xylan > two-fold xylan ~ cellulose surface > cellulose internal chains > lignin (Fig. 5a). This experimental observation dovetails with a simulation reporting 50% faster diffusion for water molecules in lignin than those bound to xylan[30]. The distinctive hydration of lignin and three-fold xylan excludes homogeneous mixing, instead suggesting proximity of separate domains: lignin nanodomains close to a well-hydrated matrix of three-fold xylan (Fig. 6). The hydrated matrix further connects the junctions of two-fold xylan and surface cellulose[31], two molecules with comparable hydration (Fig. 5a), thus bridging lignin with cellulose (Fig. 6).

Compared with primary cell walls[32], secondary walls have considerably weaker water associations, evidenced by 2–4 times slower $^1$H polarization transfer from water to macromolecules (Supplementary Fig. 6). The lignin and polysaccharide peaks in maize secondary cell walls only reached 20–30% of the equilibrium intensity within 4-ms $^1$H mixing (Supplementary Fig. 6). This water-buildup rate is notably slower than in the primary cell walls of *Arabidopsis*, which, at 4-ms, exhibits as high as 60–80% and 30–40% intensities for pectins and cellulose, respectively[32]. This can be attributed to the deposition of hydrophobic lignin, decreased water content[8], tighter packing of

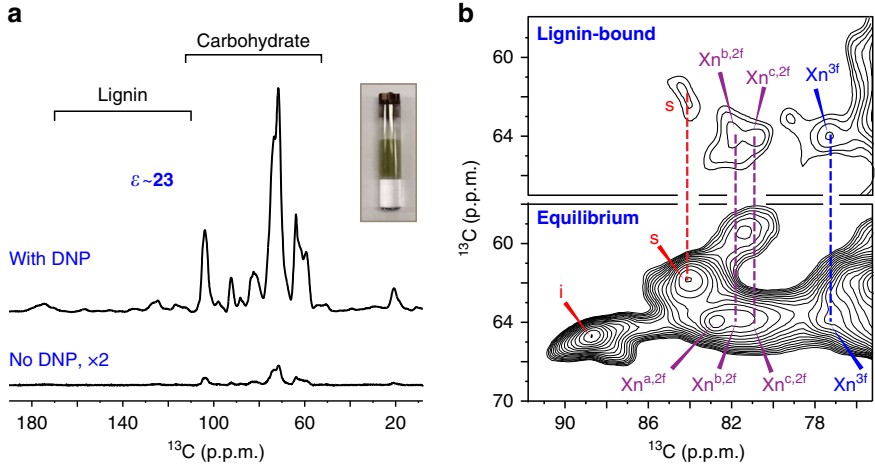

**Fig. 4** DNP reveals the conformational selectivity of xylan for lignin-binding. **a** DNP enhances the NMR sensitivity by 23-fold on maize. The inset shows a representative DNP sample. **b** Lignin-edited (top) and control (bottom) $^{13}C$–$^{13}C$ correlation spectra measured using DNP. The lignin-edited spectrum only shows polysaccharides spatially proximal to lignin, including three-fold and a subset of two-fold xylan (type b and c) and surface cellulose (s)

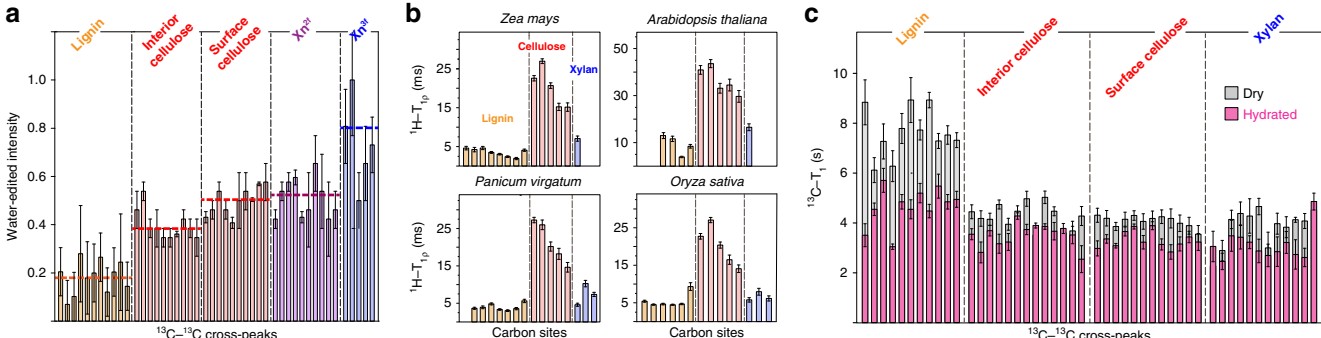

**Fig. 5** Lignin is highly hydrophobic and dynamically distinct from polysaccharides. **a** Water-edited intensities showing the hydration level of molecules in maize. Error bars are standard deviations propagated from NMR sensitivity. The hydration level decreases in the order of three-fold xylan, 2-fold xylan, surface cellulose, interior cellulose and lignin. **b** $^{1}H$-$T_{1\rho}$ relaxation times of four plants detect molecular motions on the microsecond timescale. **c** $^{13}C$-$T_1$ relaxation times of hydrated (magenta) and dried (grey) maize reflect nanosecond timescale motions. Distinct from the polysaccharide, lignin has long $^{13}C$-$T_1$ but short $^{1}H$-$T_{1\rho}$ relaxation. Error bars are standard deviations of the fit parameters. The $x$-axis corresponds to well-resolved cross peaks or carbon sites as tabulated in Supplementary Table 7–10. Source Data are provided as a Source Data file

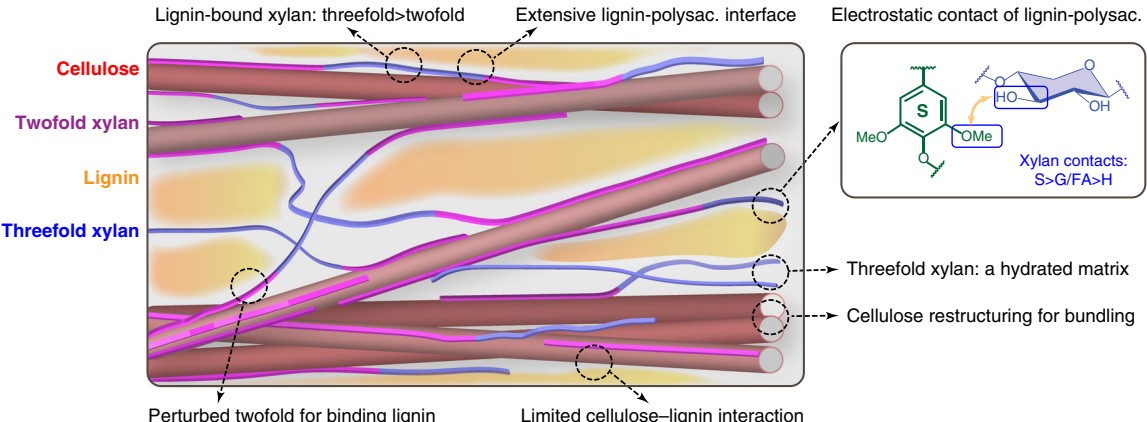

**Fig. 6** A revised model of lignin-polysaccharide packing and secondary cell wall architecture. Cellulose microfibrils, two-fold and three-fold xylan, and lignin are depicted in red, purple, blue and yellow, respectively. The secondary cell wall adopts much tighter packing than the primary cell walls, and the two hydrophobic cores of lignin and cellulose are bridged by xylans in a conformation-dependent manner. The key findings of this study have been annotated in the model, with the corresponding spots highlighted using dashline circles. The representative polar motifs responsible for lignin-xylan interactions are boxed in blue. Polysaccharides are abbreviated as polysac. The depiction may not be to scale

polymers, and lack of pectins with their water-stabilizing action. Tight packing also explains the unexpectedly large number of spatial contacts that are more prominent than those of highly porous primary cell walls[33,34].

**Lignin and polysaccharides are dynamically distinct**. The domain separation of lignin and polysaccharide is confirmed by their distinct dynamics. Lignin has the shortest $^1H$-$T_{1\rho}$ relaxation times of all wall polymers (Fig. 5b) and its fast relaxation can be attributed to μs motions similar to the ring flips of protein tyrosine or phenylalanine sidechains[35] or the collective motions of multiple rings. Similar motions are not feasible in the hydrogen-bonded glucan chains of cellulose microfibrils, which exhibit the longest $^1H$-$T_{1\rho}$ relaxation times. This trend is fully reversed for the $^{13}C$-$T_1$ relaxation, with the longest times for lignin, ~5 s for hydrated sample and 8 s for dried materials (Fig. 5c), indicating that aromatics resist rapid local reorientation on the ns timescale. In contrast, the $^{13}C$-$T_1$ relaxation is markedly shorter for all polysaccharides due to efficient spin diffusion across the tightly packed polysaccharides. These dynamical heterogeneities support the concept of inhomogeneous mixing on the nm scale. The aromatic rings exhibit pronounced μs motion, thus are unlikely to adopt the partial alignment observed in woods using Fourier-Transform Infrared (FTIR) and Raman spectroscopy[36,37]. In addition, lignin contains both rigid and mobile components (Supplementary Fig. 7, 8), indicative of the coexistence of aggregated rings and polysaccharide interfaces.

The fact that pronounced $^{13}C$-$T_1$ differences between lignin and carbohydrates still exist in dehydrated samples suggests a large domain size far beyond the reach of relayed spin diffusion, a process that reduces relaxation heterogeneity for molecules within a few nanometers. Therefore, lignin domains may be as large as a few to tens of nanometers across so that the unique characteristics of motion and hydration can be fully retained.

## Discussion

This study provides heretofore unavailable molecular evidence to define the 3D architecture of secondary cell walls. The exceptional resolution and sensitivity of ssNMR and DNP, assisted by a novel experiment for analyzing the lignin-carbohydrate interface, allow us to identify 234 intermolecular cross peaks that pinpoint the packing interactions, 325 relaxation curves that probe polymer mobilities on microsecond and millisecond timescales, and 62 site-specific hydration data (Supplementary Table 3–10). This large dataset markedly alters the paradigm for lignocellulose structure. The two hydrophobic cores of lignin and bundled cellulose are bridged by xylan in a conformation-dependent manner (Fig. 6): the dehydrated two-fold xylan with a flat-ribbon structure coats the microfibril surface[15] and the remaining, well-hydrated xylans with uneven conformations connect the lignin nanodomains via extensive interfaces, along which electrostatic interactions occur between lignin methyl ethers and sugar polar groups. This well-defined boundary between aromatics and polysaccharides validates the modeling results in which lignin and xylans tend to self-aggregate to form separate phases, between which only limited interpenetration can happen[30].

The numerous cross peaks between xylan and lignin, and between different carbon sites, mainly represent through-space, sub-nanometer contacts rather than covalent linkages. In fact, the lack of through-bond correlations between the arabinose sidechains of xylan and the ferulate lignin in the INADEQUATE spectrum (Fig. 1a, f) refines the hypothesis that xylan sidechains are crosslinked to lignin via ferulate or diferulate residues[38,39]. Such covalent bonding may only occur at limited spots, well below the detection limit of ssNMR. The abundant noncovalent

interactions documented here predominate the physical interactions between xylan and lignin.

Pectin has also been assumed to be covalently connected to lignin via ester or ether linkages and bridge the lignin to hemicellulose[40]. Pectin has been proposed to play a crucial role in lignification, especially in the initiation steps, as evidenced by biomimetic polymerization process of coniferyl alcohol in pectin solutions[41–43]. The intact stems analyzed here only contain a minor fraction of pectic substances and we have not identified any lignin-pectin cross peaks. Therefore, the proposed covalent linkages between lignin and pectin are scarce, if they exist, or only happening at the early stage of lignin deposition. These two molecules are not extensively co-localized within the sub-nanometer scale in the mature plant stems.

The structure that emerges from this ssNMR study differs substantially from contemporary views of complex lignocellulose in four aspects. First, lignin is found to bind mainly xylans rather than cellulose. For decades, lignin has been considered as the glue that connects cellulose microfibrils with hemicellulose[44]. The minor cellulose-lignin interaction observed here largely discounts previous models in which cellulose aggregates are proposed to be directly coated by lignin[10] but rather suggests that lignin and cellulose are spaced and joined by xylan. Second, the phase-separation of lignin and xylan has revised an earlier model, where these two polymers are depicted to be well-mixed via entanglements and covalent linkage[11]. Third, we have emphasized the importance of electrostatic interactions over the hydrophobic contact, which is another possible mechanism proposed by simulation[23]. Finally, we have revealed, for the first time, that distorted xylan structure favors lignin-binding. This discovery, integrated with the previous finding that flat xylan conformers bind cellulose[16], has fully revealed the structure-function relationship for xylan and resolved how this versatile hemicellulose can bridge different molecules with diverse conformational structures. These novel molecular characteristics provide the structural basis for designing more digestible crops and further optimizing the biomass degradation process to facilitate the production of biorenewable energy. Similar approaches can be applied to biomasses from other plants and organisms.

## Methods

**Isotope labeling and preparation of plant materials**. The uniformly $^{13}C$-labeled stems (97% $^{13}C$) of four plants were prepared by IsoLife (Wageningen, The Netherlands) using the following protocol. Briefly, uniformly $^{13}C$-enriched (97 atom % $^{13}C$) plant stems of maize (*Zea mays*; Age 2 months after sowing), rice (*Oryza sativa*; Age 7 months), switchgrass (*Panicum virgatum*; Age 3.5 months), and *Arabidopsis thaliana*, (ecotype Columbia-O; Age 1 months) were produced under identical growth conditions in custom-designed, air-tight, high-irradiance labelling chambers of the Experimental Soil Plant Atmosphere System. Plants were grown hydroponically under controlled environmental conditions: photosynthetic photon flux density 900 (700 for *A. thaliana*) μmol m$^{-2}$ s$^{-1}$ (top of plants), 16 h (*A. th.* 14 h) day length, day/night temperature 24/16 °C, RH 75%, in a closed atmosphere containing 97 atom% $^{13}CO_2$ ($CO_2$ enriched with the stable isotope $^{13}C$; from pressurized cylinders, Cambridge Isotope Laboratories) from germination till harvest. Minerals and water were supplied as a modified Hoagland-type nutrient solution with micronutrients[45] and iron as Fe(III)-EDTA[46], modified to contain a maximum of 25% of total N as ammonium, maintaining pH between 5 (maize: 4) and 6. At harvest, immediately after removing the plants from the growth chamber, plant shoots were dissected into leaves and stems, cut to 2–4 cm pieces, weighed, packaged in food-grade PE pouches, and stored at −30 °C. After freeze-drying, representative subsamples were prepared for $^{13}C$ analysis by high-abundance isotope ratio mass spectrometry (IRMS; Stable Isotope Facility, UC-Davis, CA, USA). $^{13}C$ abundance of $CO_2$ in the chamber atmosphere was kept close to 97 atom% $^{13}C$, continuously monitored during culture by non-dispersive infra-red detection. All the four plant samples were hydrated to ~40 wt % and 80–100 mg were sliced and packed into a 4 mm Bruker magic-angle spinning (MAS) NMR for ssNMR experiments. Around 30 mg of maize were packed into a 3.2-mm Revolution NMR rotor. Another 28 mg of maize was proceeded in DNP matrix and transferred to a 3.2-mm sapphire rotor for MAS DNP experiments as detailed below.

**ssNMR experiments for resonance assignment**. Solid-state experiments were conducted on a Bruker Avance 600 MHz (14.1 Tesla) spectrometer and a 400 MHz (9.4 Tesla) Bruker Avance spectrometer using 3.2-mm and 4-mm MAS HCN probes respectively. Most experiments except those with MAS-DNP were collected under 10–14 kHz MAS at 294–298 K. $^{13}C$ chemical shifts were externally referenced to adamantane $CH_2$ signal at 38.48 ppm on the TMS scale. Typical radiofrequency field strengths were 80–100 kHz for $^1H$ decoupling and hard pulse, 62.5 kHz for $^1H$ CP and 50–62.5 kHz for $^{13}C$.

To assign the NMR signals of polysaccharides and lignin, 2D double-quantum (DQ) correlation spectra were recorded using the refocused CP J-INADEQUATE pulse sequence[28,47], which relies upon the scalar coupling between two $^{13}C$ nuclei to obtain through-bond information regarding directly couple $^{13}C$ nuclei. A 2D $^{13}C$–$^{13}C$ correlation spectrum is also measured with 53-ms CORD mixing[48]. All the assigned $^{13}C$ chemical shifts of both polysaccharides and lignin have been summarized in Supplementary Table 2.

To analyze the lignin composition in four different plants, spectral deconvolution of the $^{13}C$ quantitative DP spectra was conducted using the software DMfit[49]. The peak area of the 159 ppm, 153 ppm and 147 ppm were assigned to the H4, S2/6 and the mixture of G3/4 and FA3/4/7, respectively (Supplementary Fig. 1c). To convert the peak area into molar percentage, the carbon numbers of each peak and the residue multiplicity need to be considered. We have thus divided the relative intensity of S2/6 by 2, and the G3/4 and FA3/4/7 peak intensity by either 2 or 3 to account for the two extreme conditions. The resulting error margin was well below 3% for the H and S residues in all plants and below 5% for the G/FA residues.

The aromatic signals of lignin are typically difficult to analyze in solids due to the signal suppression caused by large chemical shift anisotropy. Here we employ a modified version of the standard proton-driven $^{13}C$ spin diffusion (PDSD) method that reintroduces $^{13}C$-$^1H$ dipolar coupling via a gated decoupling period to enhance lignin signals against the proton-rich polysaccharides (Supplementary Fig. 3, 4)[50]. A total dipolar dephasing period of 68 µs is employed to reintroduce $^{13}C$-$^1H$ dipolar couplings that selectively suppress the signals of protonated carbons. This dipolar dephasing period is asymmetric with respect to the π pulse in the Hahn echo, containing two undecoupled delays of 46 µs and 22 µs[51]. A 100 ms mixing period is applied to both the gated-PDSD and the standard DARR experiment. The standard DARR experiment better detects the protonated carbons due to the use of CP while the gated-PDSD preferentially detects the nonprotonated carbons. Adding these two spectra with the same number of scans better presents the lignin signals due to the compensation of spectral asymmetry caused by the proton density heterogeneity in the aromatic motifs.

**ssNMR experiments for structural determination**. To determine lignin-carbohydrate interactions, 2D gated PDSD experiments were measured on all four plant samples using 100 ms mixing time for detecting intramolecular cross peaks and 1 s for long-range intermolecular cross peaks. 74, 62, 59, and 39 intermolecular cross peaks have been identified for the rice, Arabidopsis, switchgrass, and rice, respectively. These 234 long-range cross peaks were categorized into 87 strong, 68 medium and 79 weak cross peaks according to the relative area of each single cross peak within a whole 1D $^{13}C$ cross-section. For the 74 cross peaks in maize, they include 26 strong, 24 medium and 24 weak cross peaks. The intensity cutoff is set to >5.0% for strong restraints, 2.5–5.0% for medium restraints, and <2.5% for weak restraints (Supplementary Table 3–6). This strategy has been recently employed to determine the structure of fungal cell walls[52]. These structural restraints, together with the lignin-edited spectra measured using MAS-DNP (see below), provide site-specific information of the cell wall packing.

To determine the site-specific hydration of lignin and polysaccharides, we conducted the water-edited 2D $^{13}C$−$^{13}C$ correlation experiment (Supplementary Fig. 4c)[32,53], which generated 62 hydration restraints. This experiment uses a $^1H$-$T_2$ relaxation filter of 0.71 ms × 2 to suppress the polysaccharide signals to less than 2% but retains 75% of water magnetization. The water-polarization is further transferred to spatially proximal polysaccharides using a 2.25-ms $^1H$ mixing period and a 1 ms $^1H$-$^{13}C$ CP for $^{13}C$ detection. A 100-ms DARR mixing period is used for both the water-edited spectrum and the control 2D $^{13}C$–$^{13}C$ correlation spectrum showing full intensity. Both the control and water-edited 2D spectra are plotted with normalization to the worst hydrated iC4-iC6 cross peaks at (89, 65 ppm). Both spectra are plotted using a bottom level that is set to 20% of the iC4-iC6 peak height, a level increment for multiplication of 1.2 and 28 contour lines. The intensity ratio between the 2D water-edited spectrum and the control spectrum is quantified, which is further normalized by that of the Xn3f2/3–4 cross peak with the highest water-edited intensity (Supplementary Table 7). The 2D spectra are processed using QSINE window function (SSB 2.5) for the polysaccharide region and Gaussian Multiplication (LB, GB: −30, 0.03) for the lignin region. A series of 1D water-buildup curves are also measured using a $^1H$-$T_2$ relaxation filter of 1 ms × 2 and a $^1H$ mixing period that varies from 0.1 µs to 100 ms to obtain the water-to-polysaccharide/lignin buildup curves.

To systematically determine the molecular mobility, we have generated 325 relaxation curves (Supplementary Table 8–10) by measuring 1D/2D $^{13}C$ spin-lattice ($T_1$) relaxation and $^1H$ rotating-frame spin-lattice relaxation ($T_{1\rho}$) relaxation at 298 K under 10 kHz MAS on the 400-MHz spectrometer. The $^1H$-$T_{1\rho}$ is measured using a 62.5 kHz for spin-lock. The $^{13}C$-$T_1$ is measured using both the

standard inversion recovery and the z-filter versions[54]. The inversion recovery experiment is measured using a long recycle delay of 30 s, thus providing quantitative detection of all molecules. The Torchia $T_1$ measurement is CP-based, thus preferentially probes the rigid molecules. Furthermore, we have measured a series of 2D $^{13}C$–$^{13}C$ correlation experiment with a variable z-filter time to probe the $^{13}C$-$T_1$ relaxation with enhanced resolution (Supplementary Fig. 4d)[55]. For the dried sample, 9 of 2D spectra were measured using z-filter times of 0, 0.1, 0.4, 1, 2, 3.5, 5, 7.5 and 10.5 s, and the total experimental time is 89 h. For the hydrated sample, 7 spectra were measured using z-filter times of 0, 0.1, 0.4, 1, 2, 3.5 and 5.5 s, and the total experimental time is longer, 118 h, due to an increase in the number of scans. The intensity of each cross peak in the 2D spectra was quantified and plotted as relaxation curves. These data were fit using a single exponential decay function to obtain the site-specific relaxation times.

The dynamical profile of cell wall components is also examined using a series of one-dimension (1D) $^{13}C$ spectra measured using four different methods on a 600 MHz spectrometer under 14 kHz MAS. Refocused INEPT[56] were measured using a total polarization of 5.74 ms, which consists of two delays of 1.72 ms followed by two delays of 1.15 ms. 1D $^{13}C$ direct polarization (DP) spectra were measured using a 2 s recycle delay for the detection of dynamic components and a 40 s recycle delay to obtain quantitative detection of cell wall molecules. The difference spectrum was obtained by subtracting the 2 s DP spectrum from the 40 s DP spectrum with normalization by the number of scans, and this difference spectrum only reports rigid molecules. 1D $^{13}C$ cross polarization (CP) spectra were measured with 1-ms contact time, which preferentially improves the sensitivity of the rigid molecules.

**MAS-DNP sample preparation**. The maize stem sample was processed for MAS-DNP experiments[57–59]. The stock solution of AMUPol radical[60], the DNP matrix, was prepared using a solvent mixture of $d_8$-glycerol/$D_2O$/$H_2O$ (60/30/10 Vol%) and a radical concentration of 10 mM. About 28 mg of the $^{13}C$ maize sample was impregnated into 60 µL of the AMUPol solution and grinded for 15–20 min to allow the radical to penetrate through the plant cell wall. 28 mg of well-hydrated plant sample was then transferred into a 3.2-mm sapphire rotor. The NMR sensitivity has been enhanced by 23-fold with and without microwave irradiation. The lignin has shorter DNP buildup time (2.1 s) compared with polysaccharides (3.7 s), indicating a better association with the paramagnetic radicals. Three possible reasons might account for this preferential binding: (1) radicals may be better trapped in the aromatic network of lignins during the hand grinding of biomass in DNP matrix; (2) the polysaccharide cores are so tightly packed that the radicals, with the largest dimension of ~13 Å[60], cannot effectively penetrate through; 3) the lignin-coating on the carbohydrate cores substantially decrease the surface-to-volume ratio of polysaccharide complex, which also explains the decrease in polysaccharide accessibility upon lignin deposition.

**MAS-DNP experiments**. The DNP experiments were carried out on a 600 MHz/ 395 GHz MAS-DNP spectrometer using a 3.2 mm probe under 10 kHz MAS frequency. The cathode currents of the gyrotron were 120 mA and the temperature was 104 K with microwave on. The gyrotron microwave source was equipped with a shutter to program the duration of microwave irradiation during the experiments[61].

With the sensitivity enhancement from DNP, we are able to design a novel spectral editing experiment that combines dipolar and frequency filter to achieve clean selection of the aromatic signals of lignin over the signals (Supplementary Fig. 4b), the highest peak of which is 270 times higher than lignin peaks (Supplementary Fig. 3d-g). The microwave is on during the recycle delay and the CP excitation but turned off by the shutter after CP to prevent repolarization of the dominant signals. A short recycle delay of 2.7 s is also applied to further enhance the lignin signals over the polysaccharides that have a longer DNP buildup time of 3.7 s. The selected lignin polarization is transferred to polysaccharides using a 0.5-s PDSD mixing period and the signals of these lignin-proximal polysaccharides are detected as a 2D $^{13}C$–$^{13}C$ correlation spectrum with 20-ms PDSD mixing. The carrier frequency was set to 153 ppm for dipolar and frequency filters but changed to 114 ppm during the $t_1$ evolution and $t_2$ detection. A control spectrum with 20-ms PDSD is also measured for comparison. The DNP measurement time is 41 h for the lignin-edited 2D and 2 h for the control spectrum.

**Reporting Summary**. Further information on experimental design is available in the Nature Research Reporting Summary linked to this article.

## Data availability

All relevant data that support the findings of this study are available within the article and supplementary information or on reasonable request from the corresponding author. The Source Data underlying Figs. 2b, 3b–d, and 5a–c and Supplementary Figs 2a-c, 7 and 8f are provided as a Source Data file.

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

## Acknowledgements

This work was supported by National Science Foundation through NSF OIA-1833040. The authors thank Drs. Riqiang Fu and Zhehong Gan for experimental assistance, Dr. Brittney Nagle for proving maize coleoptile walls, Dr. Hui Yang for providing the cellulose model and Dr. Carol Taylor for constructive suggestions. DJC, along with Drs. Nagle and Yang, were supported as part of the Center for Lignocellulose Structure and Formation, an Energy Frontier Research Center funded by the US Department of Energy, Office of Science, Basic Energy Sciences under award no. DE-SC0001090. The National High Magnetic Field Laboratory is supported by National Science Foundation through NSF/DMR-1644779 and the State of Florida. The MAS-DNP system at NHMFL is funded in part by NIH S10 OD018519 and NSF CHE-1229170.

## Author contribution

X.K., A.K., F.M.V., and T.W. designed and conducted the NMR and MAS-DNP experiments. X.K. and A.K. prepared and optimized the DNP samples. X.K., A.K., D.J.C., and M.C.D.W. analyzed the experimental data. X.K., A.K., F.M.V., D.J.C., and T.W. wrote the manuscript.

## Additional information

**Competing interests:** The authors declare no competing interests.

