## [Peer Review File · Nature Communications]

Reviewers' comments:

Reviewer #1 (Remarks to the Author):

In this work, the authors have provided a detailed portrait of the tridimensional organization of the secondary cell wall of higher plants, namely maize, rice, *Arabidopsis thaliana*, and switchgrass. Notably, they show that lignin binds to xylan with a specific conformation, and that lignin forms a network with specific hydration properties. The amount of data obtained is outstanding, as is the quality of the experiments and data. The data analysis is thorough, as well as the conclusions drawn from this work. This manuscript could be published as is. This work is certainly of great interest for biophysicists and NMR spectroscopists.

Reviewer #2 (Remarks to the Author):

The manuscript by Kang et al., describes the lignin-polysaccharide interactions study in maize, rice, switchgrass and *Arabidopsis* cell walls using solid state NMR. It is very detailed investigation to understand the nature of lignin-polysaccharide interactions in secondary walls using several advanced approaches and collecting massive NMR data. The knowledge about detailed molecular organization of the plant cell wall is still the subject of intense re-investigation and additional arguments due to vast novel data collected with most advanced techniques. Presented in this manuscript results add valuable information that support the authors' model depicting the distribution of lignin as nanodomains within secondary cell walls without substantial mixing with polysaccharides via covalent interactions. This study provides new information about lignin organization in plant secondary cell walls and, therefore, has strong merits for publication. However, there are few points that need to be addressed before this manuscript can be accepted.

Main text and figures:

Page 3, line 44-45: what does it mean "...limit molecular support for this organization"? Do authors mean there is no much information about cell wall organization on molecular level? This statement has to be clearer.

Page 4, line 64-65: I do not think that the results obtained really tell us about biosynthesis and assembly of secondary cell wall. They do not offer the answer to how the lignin polymerization is initiated and what are actual point of initiation. Yes they demonstrated that lignin has larger surface interaction with xylan than with cellulose, but it understandable considering extensive surface of interaction between xylan and cellulose. The most valuable is the demonstration that there is no covalent interaction between lignin and polysaccharides, if it is correct, and that lignin interacts primarily with 3-folded xylan structures, although I am not sure about statement that this distorted xylan structure is indeed a prerequisite for lignin binding. I am not sure about the sequence of events, what is first and what is second, first xylan gets distorted or interaction with lignin induce such distortion. I did not find direct statement about covalent interactions, but such conclusion, I guess, can be drawn indirectly from all observations reported in this work.

Page 7, line 127-128: it is stated that "methyl carbon of xylan acetyl groups with the carbon 9 of ferulate and the carbon 3/5 of syringyl units in lignin", which seems the most abandon interaction site between lignin and xylan, but in the final figure 6 in the window on the left the arrow indicate interaction between the carbon in xylan ring and Me group on syringyl unit. So what is the correct site of interaction?

Page 8, line 150-151: il guess there is a mistake in the statement because S-residue is in lignin, so how it can form cross peaks with lignin?

Page 9, line 174: The beginning of sentence has to be corrected. In the previous sentence they talk about 3-fold conformation of xylan and then in the following sentence they start with "These 2-fold xylans..." So do they mean to continue to talk about 3-fold xylan or switched to 2-fold xylan, then word "these" confuses the reader.

Page 13, line 247: what are the "novel NMR experiments"? It would be useful to mention them here, because to my understanding the approached used in this study were used before, actually

by the corresponding author himself, or maybe I am missing something.

Line 255: stated that "...along which electrostatic interactions occur between lignin methyl ethers and sugar polar groups". But again in the figure 6, the arrow directed to carbon not even to hydroxyl of the xylose.

Line 257: I do not quite understand what authors want state in "...and only limited interpenetration can happen". Interpenetration where?

Line 260-261: The paper cited here described wood tissues, where the amount of hemicelluloses is significantly lower than in grasses, so, it is intuitively expected that surfaces of microfibrils will be more exposed for interaction with lignin, because there is not enough xylan to cover those surfaces. In this respect, there also a strong notion existed that in dicots, who have higher amount of pectins, the points of interaction/initiation of lignin are pectins not hemicelluloses (see review of Hao and Mohnen, 2014 and citations there). In this study Arabidopsis was used as the example of dicot tissues, I wonder if authors detected any pectin related peaks on their NMR spectra and were any detectible cross peaks between pectin polysaccharides and lignin. Arabidopsis does not have much secondary cell walls, therefore I guess they actually used the mixture of primary and secondary cell walls, particularly if they used also leaves.

In Supplementary document:

1. Add the note in the legend that in Arabidopsis G/FA means only G similar how it was pointed out in the main figures.
2. I could not find explanation of abbreviations for Ar1, HGA and GA. I assume these are arabinose, and glucuronic acid? So does xylan in Arabidopsis cell wall have arabinose linked to the backbone? Or the peak 108 indicates something else?

Response to Reviewers' Comments

Reviewer #1

In this work, the authors have provided a detailed portrait of the tridimensional organization of the secondary cell wall of higher plants, namely maize, rice, *Arabidopsis thaliana*, and switchgrass. Notably, they show that lignin binds to xylan with a specific conformation, and that lignin forms a network with specific hydration properties. The amount of data obtained is outstanding, as is the quality of the experiments and data. The data analysis is thorough, as well as the conclusions drawn from this work. This manuscript could be published as is. This work is certainly of great interest for biophysicists and NMR spectroscopists.

We thank reviewer for the positive comments regarding our study. Yes, this study provides consolidated, molecular evidence on the mechanism of carbohydrate-lignin binding and advances our understanding of plant cell walls and biomass.

Reviewer #2

The manuscript by Kang et al., describes the lignin-polysaccharide interactions study in maize, rice, switchgrass and *Arabidopsis* cell walls using solid state NMR. It is very detailed investigation to understand the nature of lignin-polysaccharide interactions in secondary walls using several advanced approaches and collecting massive NMR data. The knowledge about detailed molecular organization of the plant cell wall is still the subject of intense re-investigation and additional arguments due to vast novel data collected with most advanced techniques. Presented in this manuscript results add valuable information that support the authors' model depicting the distribution of lignin as nanodomains within secondary cell walls without substantial mixing with polysaccharides via covalent interactions. This study provides new information about lignin organization in plant secondary cell walls and, therefore, has strong merits for publication. However, there are few points that need to be addressed before this manuscript can be accepted.

We thank reviewer for acknowledging the soundness, novelty and broad interest of this study. We have address all of the comments in a point-to-point manner, which, we believe, has substantially improved the clarity of this manuscript.

Main text and figures:

Page 3, line 44-45: what does it mean "...limit molecular support for this organization"? Do authors mean there is no much information about cell wall organization on molecular level? This statement has to be clearer.

Yes, this is exactly what we mean. We have rewritten that sentence for clarity: "However, due to the inherent, technical constraints of traditional analytical methods, detailed molecular information about secondary cell wall organization has remained scarce."

Page 4, line 64-65: I do not think that the results obtained really tell us about biosynthesis and assembly of secondary cell wall. They do not offer the answer to how the lignin polymerization is initiated and what are actual point of initiation. Yes they demonstrated that lignin has larger surface interaction with xylan than with cellulose, but it understandable considering extensive surface of interaction between xylan and cellulose.

We have revised this sentence to emphasize the advance in “the supramolecular architecture of secondary plant cell walls”, which focuses more on the structural side rather than the biosynthesis.

The most valuable is the demonstration that there is no covalent interaction between lignin and polysaccharides, if it is correct, and that lignin interacts primarily with 3-folded xylan structures, although I am not sure about statement that this distorted xylan structure is indeed a prerequisite for lignin binding. I am not sure about the sequence of events, what is first and what is second, first xylan gets distorted or interaction with lignin induce such distortion. I did not find direct statement about covalent interactions, but such conclusion, I guess, can be drawn indirectly from all observations reported in this work.

As our method is mainly determining the spatial packing of biomolecules, these data mainly tell us how lignin and carbohydrates are packed in native plant stems. It can be drawn indirectly from all our observations that the covalent linkages, if present, are relatively minor in population. Also, we cannot identify any clear through-bond cross peaks between xylan and lignin, which confirms that the putative covalent interactions are either absent or below the detection limit. We have added two new paragraphs in the discussion section to clarify the nature of intermolecular cross peaks and the lack of evidence for covalent linkages in our samples.

The averaged structure shows that distorted xylan binds lignin better but the sequence of events is unclear, which may require many future studies that combine plant biology with spectroscopy to answer this question. We have now revised this statement on Page 14 to only focus on the structural side: “...distorted xylan structure favors lignin-binding.”

Page 7, line 127-128: it is stated that “methyl carbon of xylan acetyl groups with the carbon 9 of ferulate and the carbon 3/5 of syringyl units in lignin”, which seems the most abundant interaction site between lignin and xylan, but in the final figure 6 in the window on the left the arrow indicate interaction between the carbon in xylan ring and Me group on syringyl unit. So what is the correct site of interaction?

These two cross peaks are just examples rather than the most abundant interaction site. We have now substantially expanded this paragraph to classify the intermolecular contacts into four categories and provide more examples and explanations. Since we are studying the complex and intact stem of plants, the cross peaks can be found between

almost all carbons in lignin ring and methyl groups with xylan C1-C5 and acetyl groups. (Fig. 2, Supplementary Fig. 2 and Tables 3-6). Therefore, it is not straightforward to define a single site of interaction between lignin and xylan. However, a statistical view of the identified xylan-syringyl contacts did reveal that the strongest interaction site happens between syringyl methyl ether groups and the xylan polar groups, which is shown in the revised Figure 6.

Page 8, line 150-151: I guess there is a mistake in the statement because S-residue is in lignin, so how it can form cross peaks with lignin?

Thanks for pointing out this mistake. It meant to be “cross peaks with xylan.” We have corrected it.

Page 9, line 174: The beginning of sentence has to be corrected. In the previous sentence they talk about 3-fold conformation of xylan and then in the following sentence they start with “These 2-fold xylans....” So do they mean to continue to talk about 3-fold xylan or switched to 2-fold xylan, then word “these” confuses the reader.

We switched from the 3-fold to 2-fold xylan. We have now rewritten this sentence to emphasize the transition: “In contrast, the 2-fold xylans are”

Page 13, line 247: what are the “novel NMR experiments”? It would be useful to mention them here, because to my understanding the approaches used in this study were used before, actually by the corresponding author himself, or maybe I am missing something.

The novelty resides in the DNP-assisted, lignin edited experiment (Fig. 4), which does not exist before and is specifically designed to probe the lignin-carbohydrate interface. We put all the NMR experimental details to the Methods section and the new NMR pulse program to Supplementary Fig. 4b to improve the readability for non-NMR readers. We have now clarified the novelty of this experiment as below.

Maintext: “The exceptional resolution and sensitivity of ssNMR and DNP, assisted by a novel experiment for analyzing the lignin-carbohydrate interface, allow us ...”

Methods: “...we are able to design a novel spectral editing experiment that combines dipolar and frequency filter to achieve clean selection of the aromatic signals of lignin over the signals.”

Line 255: stated that “...along which electrostatic interactions occur between lignin methyl ethers and sugar polar groups”. But again in the figure 6, the arrow directed to carbon not even to hydroxyl of the xylose.

The previous emphasis on the carbon sites for the illustrative figure since the NMR data is based on carbon-carbon correlations. We have now revised figure 6 to highlight the

hydroxyl groups in xylan and methyl ethers in lignin, which conveys the information much better.

Line 257: I do not quite understand what authors want state in "...and only limited interpenetration can happen". Interpenetration where?

We have rewritten this sentence to improve the clarity. The modeling results indicate that lignin and xylan tend to phase-separate, and only very limited interpenetration between these two domains can occur.

Line 260-261: The paper cited here described wood tissues, where the amount of hemicelluloses is significantly lower than in grasses, so, it is intuitively expected that surfaces of microfibrils will be more exposed for interaction with lignin, because there is no enough xylan to cover those surfaces. In this respect, there also a strong notion existed that in dicots, who have higher amount of pectins, the points of interaction/initiation of lignin are pectins not hemicelluloses (see review of Hao and Mohnen, 2014 and citations there). In this study *Arabidopsis* was used as the example of dicot tissues, I wonder if authors detected any pectin related peaks on their NMR spectra and were any detectible cross peaks between pectin polysaccharides and lignin. *Arabidopsis* does not have much secondary cell walls, therefore I guess they actually used the mixture of primary and secondary cell walls, particularly if they used also leaves.

In order to concentrate the abundance of secondary cell walls, we are using the mature stems (without leaves). We have screened many pieces of such stems until a satisfactory sample (with mainly secondary cell wall components) was identified. Among the four plant species we examined, *Arabidopsis* clearly has the most pectin (Fig. a below) but there is no pectin-lignin cross peak in the long-range 2D correlation spectra (Fig. b).

We have now added a new paragraph to the discussion section to explain the phase-separation of lignin and pectin. Six relevant papers are also added to the references.

In Supplementary document:

1. Add the note in the legend that in Arabidopsis G/FA means only G similar how it was pointed out in the main figures.

We have now updated this annotation in the Supplemental Figure 2a and the statement in the figure legend.

2. I could not find explanation of abbreviations for Ar1, HGA and GA. I assume these are arabinose, and glucuronic acid? So does xylan in Arabidopsis cell wall have arabinose linked to the backbone? Or the peak 108 indicates something else?

Yes, Ar, HGA, and GA are abbreviations for arabinose, homogalacturonan and galacturonic acid, respectively. These arabinose signals can be from both xylan and pectin sidechains. We have now updated the assignments and table legends in the Supplementary Table 8 and 9 to keep the consistency with previous study (Wang et.al., Biochemistry 2014).

REVIEWERS' COMMENTS:

Reviewer #2 (Remarks to the Author):

All my concerns and comments were addressed. My suggestion to accept the manuscript for publications